# Y chromosome sequence and epigenomic reconstruction across human populations

Paula Esteller-Cucala [1,7 ✉], Marc Palmada-Flores[1,7], Lukas F. K. Kuderna[1], Claudia Fontsere[1], Aitor Serres-Armero[1], Marc Dabad [2], María Torralvo [1], Armida Faella[1], Luis Ferrández-Peral[1], Laia Llovera[1], Oscar Fornas [3,4], Eva Julià [3], Erika Ramírez[3], Irene González[3], Jochen Hecht[3], Esther Lizano [1,5], David Juan[1] & Tomàs Marquès-Bonet [1,2,4,5,6 ✉]

Recent advances in long-read sequencing technologies have allowed the generation and curation of more complete genome assemblies, enabling the analysis of traditionally neglected chromosomes, such as the human Y chromosome (chrY). Native DNA was sequenced on a MinION Oxford Nanopore Technologies sequencing device to generate genome assemblies for seven major chrY human haplogroups. We analyzed and compared the chrY enrichment of sequencing data obtained using two different selective sequencing approaches: adaptive sampling and flow cytometry chromosome sorting. We show that adaptive sampling can produce data to create assemblies comparable to chromosome sorting while being a less expensive and time-consuming technique. We also assessed haplogroup-specific structural variants, which would be otherwise difficult to study using short-read sequencing data only. Finally, we took advantage of this technology to detect and profile epigenetic modifications among the considered haplogroups. Altogether, we provide a framework to study complex genomic regions with a simple, fast, and affordable methodology that could be applied to larger population genomics datasets.

[1] Institut de Biologia Evolutiva (CSIC-Universitat Pompeu Fabra), Doctor Aiguader 88, Barcelona, Spain. [2] CNAG-CRG, Centre for Genomic Regulation (CRG), Barcelona Institute of Science and Technology (BIST), Baldiri i Reixac 4, Barcelona, Spain. [3] Centre for Genomic Regulation (CRG), Barcelona Institute for Science and Technology (BIST), Doctor Aiguader 88, Barcelona, Spain. [4] Universitat Pompeu Fabra (UPF), Doctor Aiguader 88, Barcelona, Spain. [5] Institut Català de Paleontologia Miquel Crusafont, Universitat Autònoma de Barcelona, Edifici ICTA-ICP, Cerdanyola del Vallès, Spain. [6] Institució Catalana de Recerca i Estudis Avançats (ICREA), Passeig Lluís Companys 23, Barcelona, Spain. [7] These authors contributed equally: Paula Esteller-Cucala, Marc Palmada-Flores. ✉email: paula.esteller@upf.edu; tomas.marques@upf.edu

Human sex chromosomes have been traditionally excluded from genome-wide studies[1,2]. This exclusion is particularly pronounced for the Y chromosome, the study of which could be key in understanding differences in disease susceptibility between men and women[3–5]. However, the Y chromosome is now considered important not only for male-specific traits but also for the study and characterization of common complex diseases[4]. Sex-limited chromosomes, defined as those unique to a heterogametic genome[6], are usually harder to assemble since they are haploid and thus have half the sequencing depth when sequenced together with other autosomal and homogametic chromosomes. Moreover, their repetitive nature, filled with ampliconic regions and heterochromatin, poses an additional challenge for assemblers[7].

The first Y chromosome assemblies were generated by means of bacterial artificial chromosomes (BACs), which are labor-intensive and time-consuming approaches[8–10]. Indeed, the Y chromosome sequences in the GRCh38 assembly[11–13] are a composite of BAC clones[11] from a male that belongs to the R1b haplogroup[14] and pseudoautosomal (PAR) regions from the X-chromosome.

To facilitate the assembly process and also to avoid the use of such costly techniques, one can decrease the potential inter-chromosomal assembly overlaps by specifically enriching the chromosome of interest. This can be done by physically isolating the chromosome using flow cytometry (chromosome sorting)[6,15–17]. Alternatively, other selective sequencing methods, such as adaptive sampling on Oxford Nanopore Technologies (ONT) devices[18], can potentially be used.

Chromosome sorting allows the chromosome of interest to be sequenced on different platforms after its physical isolation by flow cytometry. This separation is possible because different chromosomes have specific fluorescence intensity[19]. On the other hand, adaptive sampling allows for the sequencing of specific DNA regions by sequence enrichment or depletion of off-target reads during sequencing without the need for previous chromosome enrichment[20,21]. Any DNA molecule that does not correspond to the genomic region of interest will be ejected from the pore, thereby preventing any further sequencing. To obtain a de novo assembly, it is also important to avoid whole-genome amplification (WGA), as this process can introduce chimeras, bias the assembly process[22], and prevent the detection of epigenetic modifications.

Long-read whole-genome sequencing enables the assessment of previously unsolved repeats and thus allows the generation of more contiguous assemblies. Currently, ONT can achieve the longest read lengths compared to any other existing sequencing technology[23–25]. Moreover, ONT allows the detection of DNA (and RNA) modifications based on the different current signals of the nanopores[26,27]. Taken together, this technology is able to resolve gaps, allowing for the true completion of chromosomes or even genomes[28–30]. Here we assess the performance of two enrichment methods to sequence and assemble the Y chromosomes from seven major human haplogroups. Moreover, we provide insights into their structural variation and epigenomic landscape, showing that enrichment techniques coupled with ONT can be used to study variation between population datasets.

## Results

**Data production.** Complete Y chromosomes from six different human haplogroups were isolated as previously described[17] (Fig. 1a, b). In brief, chromosomes were obtained from lymphoblastoid cell lines (LCLs) used in the 1000 Genomes Project[31] (1kgp) and sequenced on the ONT MinION. We also made use of the Y chromosome sorted ONT data generated by Kuderna et al.[17], whose haplogroup (A0) represents one of the deepest-rooting known haplogroups. Additionally, we also generated Illumina short-read data for the same flow-sorted chromosomes (Supplementary Data 1).

The ONT data available for the cell lines ranged from 6.4 to 10.3 Gb, of which 7–33% mapped to the Y chromosome in the reference. Moreover, we also generated 6.4–35 Gb of Illumina data for all the chromosome sorting extractions. This is a notably high amount of data, especially considering that the Y chromosome sequence represents less than 1% of the known sequence in GRCh38.

The Y chromosome enrichment specificity was assessed by aligning the basecalled data to the human reference genome assembly GRCh38 and calculating the normalized coverage on each chromosome accounting for the gaps of the reference genome and the ploidy of each chromosome ("Methods"; Fig. 1c and Supplementary Figs. 1 and 2). The Y chromosome-specific enrichment factor of the six samples showed high variability, as it ranged from 15- to 50-fold, whereas the A0 haplogroup was over 100-fold enriched (Supplementary Data 2). As noted in Kuderna et al.[17], we found that chromosome 22 partially co-sorts with chromosome Y, showing enrichment values slightly higher than 1.

**Adaptive sampling as a strategy to enrich specific chromosomes.** A limiting factor in chromosome sorting is the need to culture hundreds of millions of cells in order to enrich the chromosome of interest effectively[16,17]. To overcome this limitation, we explored the potential of adaptive sampling to specifically enrich the Y chromosome. This approach was done for one of the cell lines (haplogroup H) for which chromosome sorting data had also been generated. We used the nucleotide sequences of the Y chromosome (chrY) and the contig *chrY_KI270740v1_random* (chrY_random) as provided in the GRCh38 assembly as the target sequences to enrich. To obtain comparable coverages using the two methodologies (~18x), we ran two ONT MinION flowcells with adaptive sampling. In both experiments, we showed that the Y chromosome was preferentially enriched to the other chromosomes (Supplementary Fig. 3). Although the Y chromosome enrichment factor value with chromosome sorting doubles the one in adaptive sampling for this cell line (Fig. 1c; Supplementary Data 2), adaptive sampling proves to be a cheaper and less time-consuming strategy.

**Y chromosome assembly across haplogroups and enrichment techniques.** We obtained Y chromosome data and assemblies corresponding to seven different Y chromosome haplogroups using chromosome sorting on distinct cell lines ("Methods"; Supplementary Figs. 4–6; Supplementary Data 3). The coverage used by the assembler to generate each assembly ranged from 13 to 50x, with a mean assembly coverage of over 28x. The resulting assemblies spanned from 18.95 to 22.08 Mb in length, being 16–28% shorter than the length of chromosome Y in GRCh38. We also observed that assemblies with higher continuity (contig N50) tend to have higher values of read length N50 and mean read lengths (Supplementary Data 3).

Our assemblies had a similar amount of contigs compared to a previously published African Y chromosome assembly (haplogroup A0)[17], which is lower than the number of contigs of the GRCh38. The N50 across our assemblies ranged from 1.40–2.67 Mb and are thus within the same order of magnitude as the Y chromosome in GRCh38 (6.91 Mb). These results suggest that creating de novo assemblies primarily based on long reads mapping to a reference chromosomal assembly might lead to shorter assemblies, less fragmented but with lower continuity values (such as lower contig N50). However, these results also

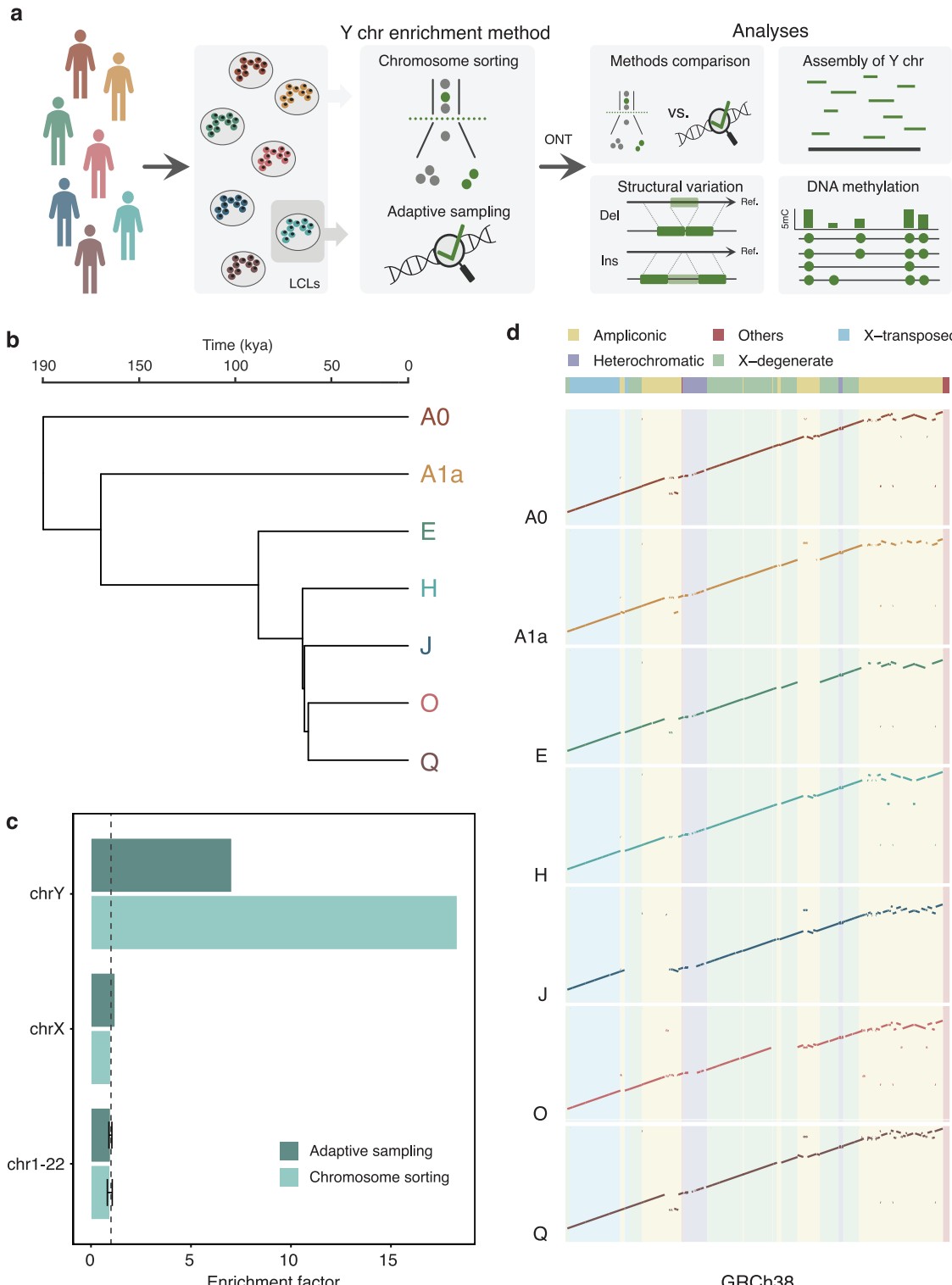

**Fig. 1 Study design, enrichment factor, and assemblies. a** Summary of the samples, methodologies, and analyses used in the study. **b** Phylogenetic tree of the human Y chromosomes used in the study. Split times are taken from Jobling and Tyler-Smith[79]. kya, kilo years ago. **c** Enrichment factor values of the H haplogroup from data generated using chromosome sorting and adaptive sampling. The chrY shows higher enrichment with chromosome sorting than adaptive sampling for the haplogroup compared. The dashed vertical line equal to 1 denotes no chromosomal enrichment. For the autosomal chromosomes, the mean enrichment value is displayed, and error bars represent the standard deviation (*n* = 22 autosomal chromosomes). **d** Dot-plots of the manually scaffolded Y chromosomes compared to the resolved MSY region of GRCh38. The large-scale deletion in the J haplogroup is most likely due to its low coverage. Source data are provided in Supplementary Data 8.

show that by using ONT and Illumina platforms, it is possible to generate assemblies almost as continuous as the current GRCh38 reference, for which much more effort and resources were devoted[11–13,32].

Furthermore, we manually scaffolded each haplogroup to create a single scaffold based on genome-to-genome alignments to the GRCh38 Y chromosome (Fig. 1d). Comparing our assemblies to the GRCh38 in the male-specific region of the Y chromosome (MSY) shows how most of the MSY sequence classes were assembled for the most part. As previously reported[17], the ampliconic region was the most fragmented and the least complete, most likely due to collapsed repeats. Of note, for the J haplogroup, we observed a big gap in the region comprising 6.7–9.3 Mb of the Y chromosome. This region includes an X-degenerate region and most of its adjacent ampliconic region. When inspecting the reads mapping to this region, we found few reads present, thus possibly explaining why we could not accurately assemble this region.

Compared to the previously assembled African chrY that also made use of chromosome sorting data, we were able to generate a longer and more contiguous assembly starting from the same raw fast5 reads (Supplementary Data 3). This demonstrates the value of combining up-to-date basecalling and assembly tools, which are constantly evolving for long-read data[33–35].

Apart from chromosome sorting (CS), data from the H haplogroup (GM21113 cell line) was also generated using adaptive sampling (AS). In order to generate and compare the assemblies between the two enrichment methods, and given the unequal amount of data generated between them (83.5 Mb difference), we restricted the comparison to assemblies generated using the same number of bases ("Methods"). The resulting assemblies showed similar values in metrics such as genome span (CS: 21.8 Mb, AS: 22.0 Mb), contig N50 Mb (CS: 2.7 Mb, AS: 2.6 Mb), and L50 (both L50 = 3 scaffolds). Moreover, the AS-based assembly led to a slightly more fragmented assembly (44 sequences) compared to the CS-based one (31 sequences) (Table 1 and Supplementary Fig. 7).

Altogether, we have generated assemblies for seven Y chromosome haplogroups with similar contiguity to previously published assemblies. Moreover, we also show that adaptive sampling can be used for generating assemblies that are comparable to those generated by chromosome sorting. Recently, a complete T2T human Y chromosome assembly was published[36]. This new chromosome assembly showed a higher proportion of repetitive elements (RE, Supplementary Data 4), most of which were satellite sequences. Namely 85% (53 Mb) of this new T2T-Y was annotated as repetitive[36], considerably more than the 31% (17.5 Mb) reported for GRCh38-Y and also the 68% (13–15 Mb) we found in the assemblies we generated in this study (Supplementary Data 4). Moreover, this assembly has resolved the ampliconic regions (Supplementary Fig. 8).

**The landscape of structural variants across the human Y chromosome phylogeny.** As expected by the nature of these data, methods to detect structural variants which make use of long reads show an overall better performance than methods based on short-read data[37]. Taking advantage of our data, we assessed the landscape of structural variants in the Y chromosome in the seven haplogroups. For that, we used two approaches: one based on long-read mapping (Sniffles[38,39]) and another based on assembly comparison (Assemblytics[40]).

First, we identified different structural variants based on how the reads align to a reference genome using Sniffles. We used chrY and the chrY_random sequence from the GRCh38 as the reference. After merging the indel calls ("Methods"), we identified 803 unique variants (801 indels), including 166 structural variants (at least 50 bp in size, Supplementary Data 5). The number of variants ranged from 103 to 536 events per haplogroup. Moreover, Sniffles detected one translocation in the H haplogroup and one duplication event shared between five haplogroups (all but A0 and A1a, which are basal relative to the others). The detected duplication is located in the position chrY:56,673,215, at the end of a gap. This indicates that the reference is missing a region of around 98,295 bp, similar to the sequence close to the gap. Most of the events were indels of 10–50 bp in size (Fig. 2a). Out of the 801 variants found, there were 320 insertions and 481 deletions (including three insertions and nine deletions from chrY_random). We then analyzed the repeat content of these variants and found that all indels from chrY_random consisted of LTR12B-like elements (from LTR retrotransposons type ERV1) (Supplementary Data 6). In the Y chromosome, we found 116 insertions and 153 deletions with repetitive elements (RE), and in the majority of these cases, the variants were almost entirely spanned by repeats (Supplementary Data 6).

Next, we manually investigated the longest events detected using Sniffles and confirmed the longest deletion of 6314 bp and longest insertion of 6023 bp. Both were found in haplogroup A0 and belonged to different X-degenerate regions (Supplementary Figs. 9 and 10). The longest deletion had only 38% repeat content but included different types of repeats, such as ALU-like elements (AluSc and AluSx), L1-like elements (L1PA13), and Simple Repeats (CCTTn). The repeat content of the longest insertion (98.7% of the sequence) only comprised one single LINE L1-like element, L1HS, which is one of the few retrotransposition-competent human-specific L1s[41,42] (Supplementary Data 6).

Even though the most common RE in indels are simple repeats for both insertions and deletions, proportionally more insertions were caused by smaller REs (for example, ALUs or Satellite-like), and more deletions were detected in regions with long terminal repeats (LTRs), mostly ERV1 LTR12B-like elements (Supplementary Data 6). Interestingly, regions enriched with repeat arrays harboring LTR12B motifs have been previously reported as CNV hotspots[43].

After merging and regenotyping the panel of indels, we observed that the cell line belonging to haplogroup J was the one having more undetermined genotypes (i.e., positions with no genotype information). This correlates with a lower sequencing depth for this sample. We also observed that 14% of the variants genotyped in all haplogroups shared the same genotype, which was different from the reference (Fig. 2b). Haplogroups A0 and

**Table 1 Y chromosome assembly statistics in different enrichment methods.**

| Selective sequencing method to generate the long-read data | Assembly span (bp) | Scaffold N50 (bp) | Scaffold L50 | Number of sequences |
|---|---|---|---|---|
| Adaptive sampling (AS) | 21,955,745 | 2,612,207 | 3 | 44 |
| Chromosome sorting (CS) | 21,794,102 | 2,666,112 | 3 | 31 |
| AS+CS | 22,007,578 | 2,640,901 | 3 | 42 |

Assembly metrics of Y chromosome assemblies for the GM21113 cell line (haplogroup H) using adaptive sampling and chromosome sorting, and an assembly using all the data available for GM21113.

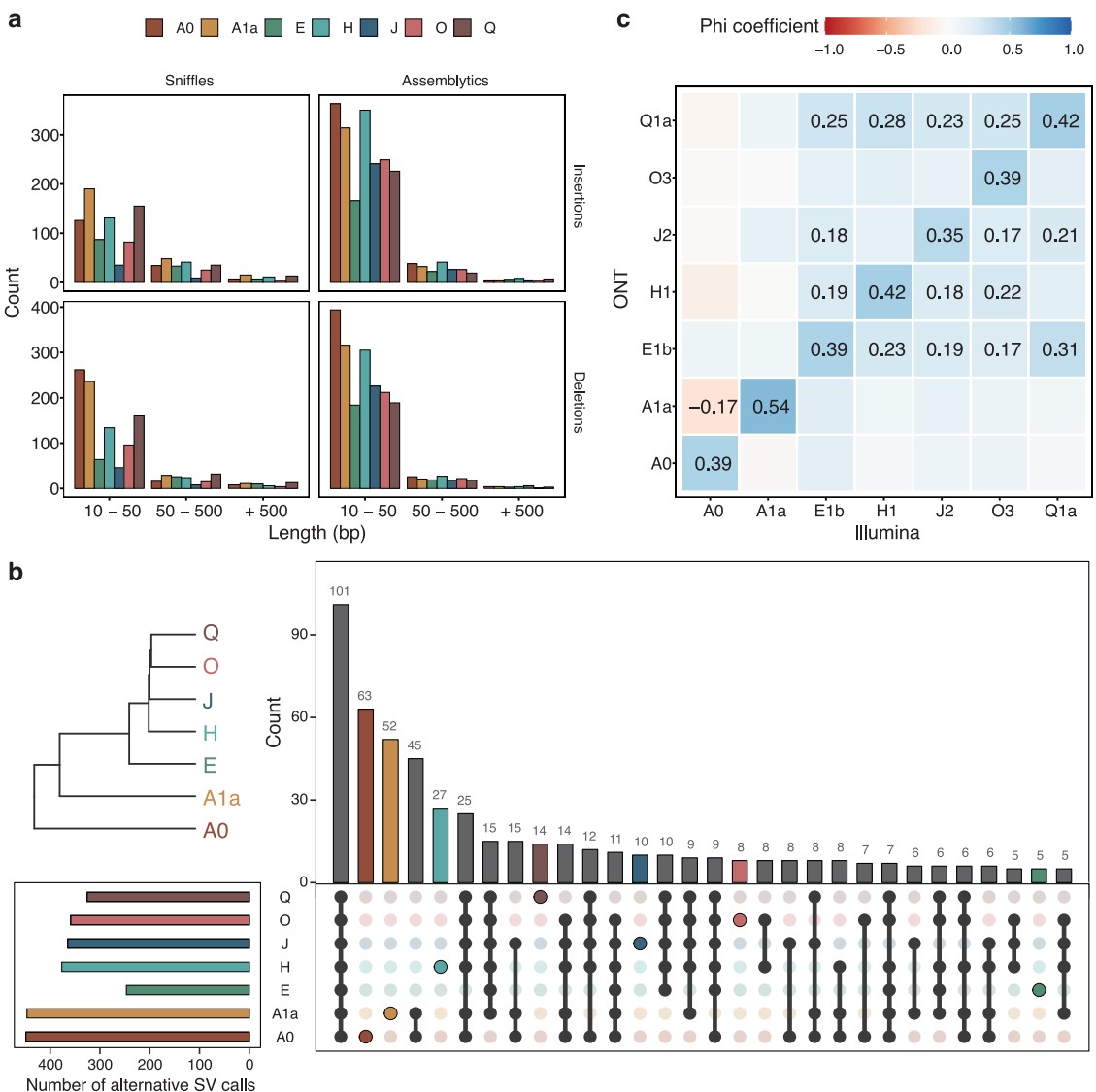

**Fig. 2 Profiling of structural variants. a** Number of the structural variant events, insertions, and deletions called by *Sniffles* or *Assemblytics* for the different haplogroups. Variants are grouped into three categories depending on their length: from 10 up to 50 bp, from 50 up to 500 bp, and equal to or over 500 bp. **b** Overlap on the alternative calls between haplogroups. As expected by their evolutionary distance, haplogroups A0 and A1a show higher haplogroup-specific variants. Only variants with genotype calls for all haplogroups have been included (*n* = 726 variants). **c** Correlations between genotype calls using *Sniffles* (ONT-based) or *graphtyper* (Illumina-based) when calling the same set of structural variants. Phi coefficients range from 1 to −1, where 1 indicates complete association. Only correlation values that are statistically significant (*p*-value < 0.05) after Bonferroni multiple testing correction are shown. Source data are provided in Supplementary Data 8.

A1a harbor the most haplogroup-specific variants, concordant to their genetic distance to the reference. We also manually assessed previously reported events for the A0 haplogroup[17] and confirmed that they were restricted to this cell line (Supplementary Figs. 11 and 12). This indicates that structural variants found in only one haplogroup might not be representative of widespread structural variants of a chromosome but rather delimited to one specific population or group of individuals.

Second, we identified structural variants based on the comparison of the obtained chrY assemblies to the reference chrY GRCh38 using *Assemblytics v1.2.1*[40] (Supplementary Data 7). This allowed for the detection of 557 to 1019 putative variants, for which 202 to 258 were at least 50 bp in size (Fig. 2a). We also found between 1 and 4 structural variants bigger than 50,000 bp for the cell lines studied, a type of variant that the mapping-based method may not detect because it would require constant coverage along a long region of the reference.

We observe 194 to 406 insertions per haplogroup with *Assemblytics*[40] compared to the 45 to 255 insertions detected with *Sniffles*[38,39], and 206 to 424 deletions against 57 to 288, respectively. However, similar amounts of structural variant indels are detected by *Sniffles* in all haplogroups (between 52 and 110) but for the J haplogroup (21) compared to *Assemblytics* (between 47 and 80, J haplogroup having 55 structural variants). These results, together with the fact that the J haplogroup is the one with less data generated, suggest that mapping-based structural variation detection methods may not be able to detect as many structural variants compared to assembly-to-assembly comparison-based methods when having limited sequencing depth. In that situation, generating a de novo assembly and using *Assemblytics* can lead to the identification of larger indels. Moreover, *Assemblytics* categorizes structural variants into multiple types including indels and other specific variants, such as tandem and repeat expansions or contractions, while *Sniffles*

was only able to capture one duplication event and one translocation, besides indels.

We further assess if the indels found using long reads could be similarly genotyped using short-read data. For that, we genotyped the variants confidently called by the ONT mapping-based approach using Illumina data generated for the same cell line extractions. With the Illumina data, we were able to replicate over one-quarter of the indels found in the nanopore data (214 out of the 801 indels). A significant positive association was seen between the predicted genotypes using ONT to those observed using Illumina data for each cell line (Fig. 2c). The observed phi coefficients (correlation values for binary variables) range between 0.35 and 0.54, not close to the highest correlation value of one[44,45]. This is expected, given that most of the variants in our panel cannot be called by the Illumina data. The intra-haplogroup correlation is generally higher than that inter-haplogroup. However, when testing inter-haplogroup associations for genotypes derived from the same methodology, strong associations were also detected (Supplementary Fig. 13). To explore which of the indels could be observed in a panel of human variation, we genotyped the same indels in the male samples present in the 1000 Genomes Project ("Methods"). As expected, intra-haplogroup associations were typically positive and significant, generally having stronger associations than inter-haplogroup comparisons, while having a limited correlation due to the fact of comparing Illumina vs ONT calls. (Supplementary Fig. 13). The use of additional cell line replicates for each haplogroup might have yielded higher associations, as it would have helped discard instances of variation that arose during immortalization.

**CpG methylation across the phylogeny**. ONT sequencing relies on the identification of different current signals when the DNA passes through the pore, so it is possible to go beyond the identification of the four canonical nucleotide bases and detect other modifications in the DNA. As such, we studied the 5mC landscape in the seven Y chromosome haplogroups. We used *nanopolish v1.12*[46] to call the methylation status of 5-methylcytosines (5mC) at CpG positions from the nanopore current signal. Assessing the Y chromosome methylome using long reads is beneficial for exploring regions that are traditionally inaccessible using short-read techniques, such as the PAR, X-transposed regions, and even the ampliconic regions.

For that, we performed quantile normalization on the methylation values across samples with a minimum coverage of 4x (Supplementary Fig. 14). We observed consistent methylation patterns along Y chromosomes across samples, indicating a strong overall correlation on the methylation status (Fig. 3a and Supplementary Figs. 15 and 16). These results alone already demonstrate the potential of ONT sequencing in studying DNA methylation in challenging genomic regions. However, 5mC frequency values could not recapitulate the expected phylogeny, either chromosome-wise or segregating by sequence class or epigenetic annotation (Supplementary Fig. 17). Given that methylation levels might vary within the population, age, environmental exposures, and cell culture conditions[47–49], and the absence of replicates for each of the haplogroups considered, this observation could be due to differences in any of these variables. However, given the uncertainties about the cell lines and age of the individuals from which they were generated, we are unable to discern the 5mC variation, which accounts for the different haplogroups from that which could be caused by other factors. As expected by the nature of the sequence classes[11], the X-degenerate region, which harbors single-copy genes and mostly ubiquitous expression, showed 5mC frequency values which resembled most of those normally seen in mammalian autosomal

chromosomes[50] (Supplementary Fig. 18). X-degenerate regions showed the characteristic bimodal distribution of frequency values with a median close to 0.7, whereas all other regions showed much less defined distributions. We also inspected the behavior of methylation according to the epigenetic annotation of the CpG of each of the sequence classes. For that, we divided the CpGs into four mutually exclusive categories (Fig. 3b and Supplementary Figs. 19 and 20): those in CpG islands (CGI), CpG shores, CpG shelves, and other inter-CGI regions (open sea). CGI in the X-degenerate and X-transposed regions were predominantly unmethylated, while all the other regions were mostly methylated. Open sea regions showed intermediate methylation levels for all sequence classes but the X-degenerate, whose median 5mC frequency reached 0.75. As expected by the dynamic nature of the human methylome, CpG shores and shelves showed intermediate values transitioning from CGI and open sea regions (Supplementary Figs. 21–23)[51–53].

DNA methylation is associated with gene expression[54], and in addition to the epigenetic annotation, we also inspected the 5mC frequency patterns across different gene annotations (Supplementary Figs. 24 and 25). Most annotated genes are present in the X-degenerate and ampliconic sequence classes (Supplementary Fig. 26), and consistent with the different expression profiles of the genes in LCLs (retrieved from GTEx[55]) in each of these two sequence classes, we observed clear distinct methylation patterns in their TSS, UTRs, and intragenic CpGs (Fig. 3c). Not surprisingly, we found 5'UTRs to be the most constrained gene feature across samples, which would directly link its methylation status to gene expression (Supplementary Fig. 27). Moreover, we found a direct relationship between upstream CGI methylation status with gene expression (Supplementary Fig. 26). Finally, we explored those cases in which differential methylation could have an effect on gene expression. We encountered a region with high methylation dispersion fully spanning a protein-coding gene (Fig. 3d and Supplementary Fig. 28). In that location, haplogroup A1a was found to be undermethylated compared to the other haplogroups, and although this difference was only modest, it could potentially modulate the expression levels of the gene located in this region. This gene is *NLGN4Y*, which is a long gene that spans over 300 kb and is expressed in the brain and other tissues, including LCLs ($\tau_{NLGN4Y} = 0.714$). Interestingly, this gene has been proposed as a candidate for autism spectrum disorder[56,57]. As expected, we found CGIs located upstream of this gene to be unmethylated (CGI_1 and CGI_2), which would be consistent with the expression of this gene in LCLs. On the contrary, a CGI potentially regulating an overlapping non-coding gene in the opposite strand and with no expression in LCLs was shown to be fully methylated in all cell lines (CGI_3). Experimental validation of the expression of these genes could ultimately confirm whether such differences in methylation truly affect gene expression levels or are merely coincidental.

Altogether, we show that ONT can be used to study 5mC across different cell lines, as it consistently recapitulates the main methylation patterns observed across mammalian genomes. Moreover, we also show that it can be helpful for the study of traditionally challenging genomic regions, particularly those present in the Y chromosome, although future work including replicates will clarify the actual sensitivity of this approach to detect differentially methylated regions.

## Discussion

Here, we present a panel of ONT data for seven cell lines that represent the major human Y chromosome haplogroups. We have generated assemblies for each of them and studied their diversity, focusing on structural variation and methylation. To

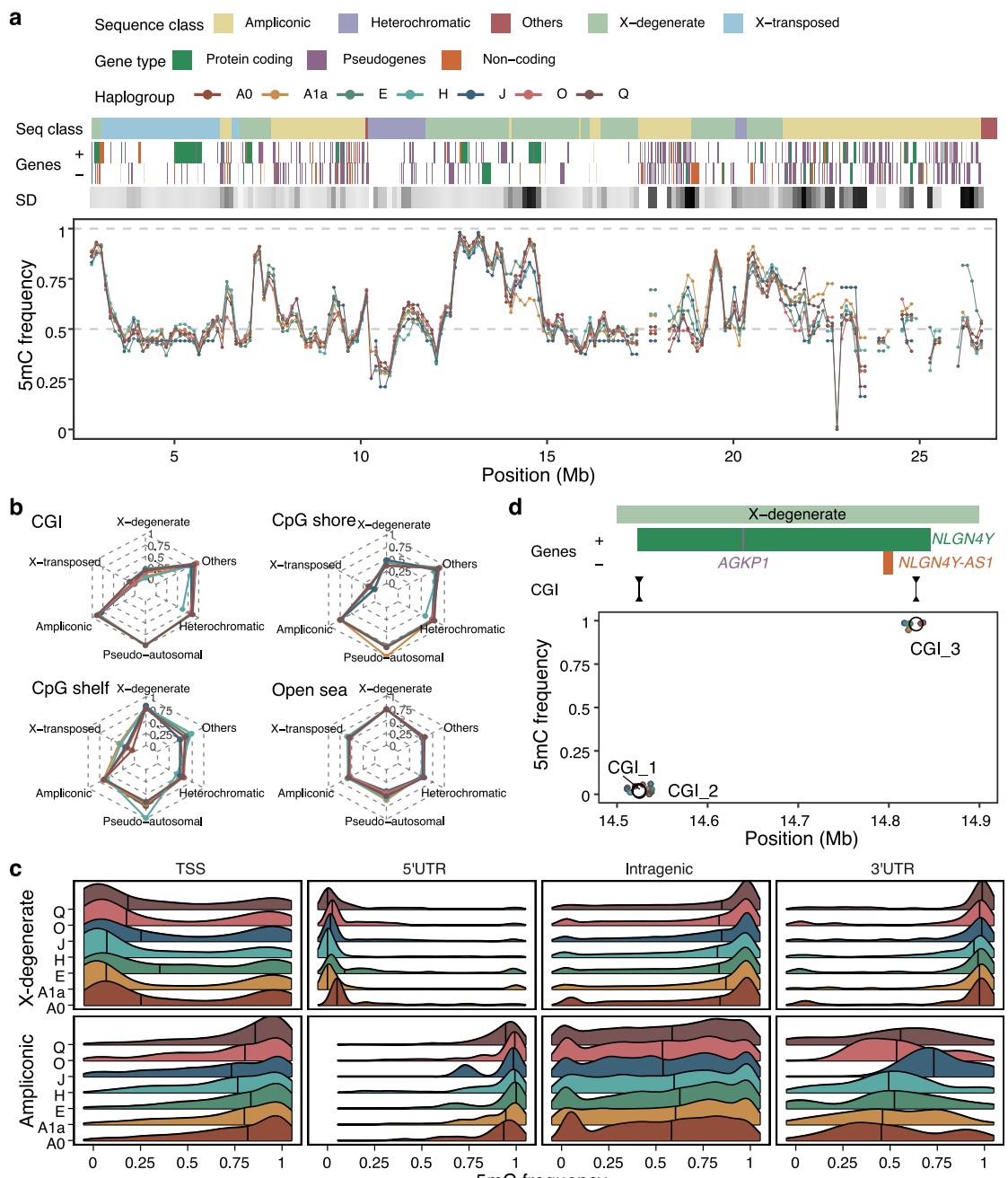

**Fig. 3 Methylation landscape across the Y chromosome phylogeny. a** Frequency of 5mC in the seven cell lines along the resolved MSY of the GRCh38. The methylation levels are calculated as the median 5mC frequency value in 250 kb sliding windows for each cell line. The sequence classes, the genes annotated, and the standard deviation (SD) of the methylation levels across cell lines are also shown. The standard deviation of the 5mC frequency is represented in a white-to-black scale, in which a darker color denotes a higher standard deviation value. **b** Median methylation value per cell line segregated by CpG annotation and sequence classes. CpG annotations are mutually exclusive regions that comprise: CpG islands (CGI), CpG shores (up to 2 kb away from the end of the CGI), CpG shelves (up to 2 kb away from the end of the CpG shores), and inter-CGI or open sea regions (where all remaining CpG are allocated). **c** 5mC frequencies on different gene features in X-degenerate and ampliconic sequence classes. Gene annotation features shown are TSS (region of 200 bp upstream of the transcription start site), both UTRs, and intragenic regions (which combine all exonic and intronic regions without considering the first gene exon). Extended version with the number of CpG sites in Supplementary Figs. 24 and 25. **d** Methylation frequencies in 3 CpG islands (CGI) surrounding the *NLGN4Y* and *NLGN4Y-AS1* genes. Empty circles show the mean 5mC frequency per CGI, whereas smaller colored points indicate the individual value in each cell line. Source data are provided in Supplementary Data 8.

generate this resource, chromosome sorting data were employed and compared to adaptive sampling data, an enrichment technique that is compatible with ONT sequencing data. After generating and comparing the assemblies of the two enrichment techniques, we showed that both methods could lead to

comparable assemblies, while they require different time, cost, and expertise. In terms of enrichment factor values, chromosome sorting shows co-enrichment with chromosome 22. This is mainly due to the fact that both chromosomes have similar sizes. However, this is not seen in adaptive sampling. In fact, samples

enriched with adaptive sampling show the lowest standard deviation of the enrichment factor on autosomal chromosomes. Nevertheless, given the homology of the sex chromosomes, and the fact that adaptive sampling is performed by providing the genomic sequence of chromosome Y, chromosome X shows a higher enrichment factor compared to the other chromosomes and the samples enriched by chromosome sorting. Altogether we show that adaptive sampling is a viable alternative strategy for the enrichment of specific genomic regions. We also emphasize the importance of using high molecular weight DNA or long DNA fragments, which are especially convenient for the enrichment of small chromosomes with adaptive sampling. As such, at longer DNA fragment sizes, the time the sequencer will be scanning for on-target regions (i.e., those that belong to the Y chromosome) will be reduced. Therefore, we realize that having started from higher DNA fragment sizes for the haplogroup H sample would have led to higher enrichment efficiencies in the adaptive sampling enrichment method.

One major limitation of our work is the conservative filtering we have used to generate the assemblies. Our approach uses the Y chromosome of the current genome of reference GRCh38 as a backbone. All data obtained using adaptive sampling relies heavily on the GRCh38 reference and may include a few reads of other chromosomes that start with a similar sequence. On the other side, chromosome sorting produces data on unresolved chromosomal regions but includes some undesired full chromosome data. As such, restricting our assembly to only those reads that map to the reference leads to the loss of a fraction of Y chromosome potentially informative reads during the filtering process. Conversely, this approach minimizes the retention of non-chromosome Y data and limits the resulting assembly to the Y chromosome only. Compared to a previous assembly created with the same data for the cell line that belongs to haplogroup A0, our approach yielded a more contiguous assembly. As such, it shows the potential that re-processing the same raw data with novel approaches might have in the future, especially in the context of the big data era[58,59].

Due to its large fraction of heterochromatin, around half of the sequence in the current Y chromosome assembly is unresolved. This limitation, together with the fact that we are using a partial reference genome to generate assemblies of a specific chromosome, hampers the possibility of reconstructing the totality of this chromosome. In the future, telomere-to-telomere Y chromosome sequencing would undoubtedly avoid reference biases we encountered in this study[28].

Due to its large fraction of heterochromatin, around half of the sequence in GRCh38 remains unresolved. This limitation, together with the fact that we are using this partial reference genome to generate assemblies of the latter, hampers the possibility of reconstructing the totality of this chromosome. The recently published telomere-to-telomere Y chromosome assembly (T2T-Y) would undoubtedly avoid reference biases we encountered in this study[28]. Our assemblies align to this T2T Y chromosome assembly in more than 89.6% of their sequence while covering more than 68% of the repetitive content. This is slightly more than what is found in GRCh38 (Supplementary Data 4). Such higher repeat content recovery showcases that reference-based approaches like the one followed in this study can include repetitive regions not present in the reference used.

We also took advantage of the long-read data generated to explore the landscape of structural variants in each cell line. For that, we used two different methods for structural variant calling: one based on long-read mapping and another based on assembly comparison. The former allows for two rounds of genotyping, and so the final candidates are potentially more curated. The latter is based on genome-to-genome comparisons, so it is able to

detect longer genomics variants. We consider that for low data samples, the creation of a de novo Y chromosome assembly may allow the detection of structural variants that cannot be recognized with a mapping-based method, considering the low coverage of reads mapping in those regions. Of note, we are aware that limited by only one replicate per haplogroup, we are not able to discriminate sample-specific variation than that appearing during the immortalization of these cells[60]. Moreover, the recent T2T Y chromosome[36] will allow for a more reliable and comprehensive assessment of the structural variants found in this study.

Besides the potential to generate high-accuracy assemblies and resolve complex genomic regions like structural variants, ONT also allows for studying the epigenome. We have assessed the methylation status of cytosines in a CpG context for our panel of cell lines. Despite the fact that the epigenome of the Y chromosome has not been deeply studied, we were able to consistently replicate the methylation patterns that have been described in other human autosomal chromosomes[61,62]. Not surprisingly, with the methylation values obtained, we were unable to recapitulate our samples' expected phylogeny. Two main factors can be attributed to this: the lack of replicates for each haplogroup and also within-population variability[48,49,63,64], which, in our case, could also be confounded by epigenetic drift[65]. Still, methylation differences at the population level are expected to be small in magnitude[48]. In this line, we were able to detect small differences in methylation in regions that could influence the regulation of specific genes. This is the case of gene *NLGN4Y*, which we found to fully overlap with a region with consistently lower methylation in the cell line belonging to haplogroup A1a. However, gene expression data from these cell lines could ultimately reveal whether such subtle variations in methylation translate into differences in gene expression. In the same line, the addition of biological replicates for different haplogroups could help uncover whether these differences are either cell-line or haplogroup-specific.

Nevertheless, we are using lymphoblastoid cell lines (LCLs), which are artificially transformed cells, so caution must be taken when extrapolating these findings. But the extent to which the generalization of our results could be biased is even more consequential when reporting those findings that are sample-specific. As such, an increase in the number of replicates would help to discern which of our findings are artifacts from those which have a true biological meaning.

Taken together, here we provide a framework to study complex genomic regions. We applied this simple, fast, and affordable technology to study diverse human population groups. Moreover, this approach can be applied to the generation of long-read data of other regions or chromosomes of interest. Long-read technologies are achieving longer and more accurate reads, which, together with higher throughputs, will enable more reliable and comprehensive comparative genomics studies, particularly for chromosomes that contain high repeat content. As such, it could be used for the characterization of virtually any species, although it would be especially advantageous for those rich in complex genomic features.

## Methods
**Flow chromosome sorting followed by ONT or Illumina sequencing**. Chromosome preparation was performed as previously described[16,17] in six lymphoblastoid cell lines (purchased from Coriell, see Supplementary Data 1 for specific details on the cell lines). The libraries to obtain the Illumina paired-end data were constructed using a SureSelect V6-Post Library Kit. Raw data generated for the haplogroup A0 (HG02982 cell line) was retrieved from Kuderna et al.[17]. The data generated in each MinION run was basecalled using *Guppy v5.0.15*[34] with the super accuracy model *dna_r94.1_450bps_sup*.

**Adaptive sampling for the enrichment of a specific chromosome**. We extracted DNA from cultured cells of haplogroup H (GM21113 cell line) using the Qiagen MagAttract HMW Kit. DNA libraries for ONT sequencing were obtained using the Ligation Sequencing Kit (SQK-LSK110) and sequenced in two ONT MinION flow-cells (FLO-MIN106 R9.4.1) using a MinION Mk1C with MinKNOW v21.02-beta4~xenial. We aimed for the specific enrichment of the chrY and the chrY_random by adding their nucleotide sequence as provided in the GRCh38 assembly. This method bioinformatically labels the reads that are being sequenced for enrichment or depletion. After a DNA strand enters the pore, the sequencer only needs one second (around 420 bases) to decide whether to continue sequencing the DNA if it matches the region of interest or to eject it if it does not. Each of the strands that enter a pore will be labeled as *unblock* and *no decision* when they are rejected in the pore or they are so short that their status remains inconclusive, respectively. They will be labeled as *stop receiving* when they are on target, thus further sequenced. Only reads labeled as *stop receiving* were used in this project.

The enrichment obtained with adaptive sampling highly depends on the fragment length of the library. Longer DNA library lengths are preferred, as the adaptive sampling enrichment algorithm takes a fixed amount of time to recognize whether to enrich a DNA strand. Because of this, in order to target a specific region of the genome that is particularly small (the Y chromosome represents ~1% of the genome), it will always be better to have few long DNA fragments rather than many short DNA fragments, as the time spent by the sequencer scanning for on-target regions will be reduced.

**Assessing the performance of two different enrichment methodologies**. The coverage and enrichment factor for each chromosome were calculated as follows:

$$\text{Coverage of chrN} = \frac{\text{Mapped bp in chrN}}{\text{Size chrN (bp, without N)}} \quad (1)$$

$$\text{Enrichment factor of chrN} = \frac{\frac{\text{Mapped bp in chrN}}{\text{Total mapped bp}}}{\frac{\text{Size chrN} \times n\,(bp,\,without\,N)}{\text{Diploid genome size}\,(bp,\,without\,N)}} \quad (2)$$

Since more than 50% of the Y chromosome in GRCh38 is composed of long stretches of unknown sequence (that in the assembly is seen as N), it is important to exclude these regions from the coverage and enrichment calculations. Because of that, Eq. 1 and Eq. 2 only consider chromosome sizes without Ns. Moreover, for calculating the enrichment and in order to account for the real target space of each chromosome, the size of each of them is multiplied by its ploidy.

**Assembly generation**. Basecalled passed reads (Q > 10) were mapped to the human GRCh38 genome assembly using *minimap2 v2.17-r941*[66] with the option *-x map-ont*. The resulting bam was indexed using *SAMTOOLS v1.12*[66,67], and the reads mapping either to chrY or chrY_random (chrY-specific reads) were retrieved.

We ran *Flye v2.9*[33] using the chrY specific reads with the option *--nano-hq* as suggested by the developers while using data basecalled using *Guppy v5*[34] onward with the super accuracy model. We added the option *--scaffold* to enable scaffolding based on the assembly graph and included two internal rounds of polishing with the argument *-i 2*.

As we used uncorrected long reads to obtain the draft assemblies, we polished the initial assemblies by first using ONT reads. We started with two rounds of *Racon v1.3.1*[68], using *minimap2 v2.9-r720*[66] with the option *-x ont* to obtain the mapping file and adding to *Racon* the argument *-u* to keep any unpolished sequences. To further improve the assembly, we then ran *medaka v1.4.1*[69] using the *medaka_consensus* program with default settings and the model *-m r941_prom_sup_g507*.

Additionally, to polish the assemblies with Illumina data, we used *HyPo v1.0.3*[70], mapping the Illumina reads to the polished assembly. For mapping short reads to the existing assembly, we used *minimap2 v2.9-r720*[66] with the option *-x sr*.

Once we polished the assemblies, we purged them using *purge_dups v1.2.5*[66,71], with default parameters and the *-2* option. This was done to remove any haplotig present in the assemblies. We obtained the mapping files using *minimap2 v2.14-r883*[66] with the option *-x ont* to map the ONT reads to the polished assembly and with the options *-xasm5 -DP* to map the split polished assembly to itself.

For the comparison of the assemblies generated from the two selective sequencing methods (chromosome sorting and adaptive sampling), we downsampled the data of the adaptive sampling experiment. For that, we used *Filtlong v0.2.0*[72] with the option *--keep_percent 87.8* so as to retrieve 87.8% of the AS sequencing data. From that point on, the assembly process was the same as the one explained (Supplementary Fig. 5).

**Genome-to-genome comparisons**. To obtain genome-to-genome alignments, we used *MuMmer v3.23*[73] *nucmer* tool with options *--maxmatch -l 100 -c 100*. To manually scaffold the Y chromosome assemblies, we used the dot-plot viewer *dot*[74]. We manually reordered and reoriented the scaffolds, joining them based on the alignments to the GRCh38 reference.

**Structural variant detection with long reads**. Structural variation was called using *Sniffles v2.0.2*[38,39] with a minimum number of reads that support an SV of *-s 10*, fed with the bam files for which we calculated and added MD tags using *SAMTOOLS v1.9*[67], with the program *samtools calmd* adding options *-uAr -Q*. We summarized the number of SVs per type and filtered out the SVs considered 'IMPRECISE' by *Sniffles*.

We merged the insertions and deletion separately with a maximum permitted distance of 100 bp (so that indels located 100 bp upward or downward will be considered a single event) found independently in all the cell lines using *SURVIVOR v1.0.7*[75]. We removed any genotype with quality under 25 (MQ) and the events that were homozygous for the reference genotype in all samples.

We used *Assemblytics v1.2.1*[40] to find structural variants in the different assemblies generated by comparing them to the reference GRCh38. We looked for structural variants with sizes between 10 and 100,000 and the unique sequence length required to call a variant of 1000.

**Indel composition**. We used *RepeatMasker v4.1.2-p1*[76] with options *-s -species human* over the different assemblies and indel reference panel to annotate repetitive sequences.

**Structural variant genotyping with short reads**. We genotyped, using the indels obtained using *Sniffles* and *SURVIVOR* as a reference, the structural variants based on Illumina data with the program *graphtyper* v2.7.5 with the option "genotype_sv" and only kept the indels with a quality >0. We genotyped them with the Illumina data generated in this study and with the Illumina data of the *1kgp*[31] available for the Y chromosome.

**Correlation between structural variant detection using long or short reads**. To assess the reproducibility of the structural variant calls obtained with ONT data in short-read data. We took the indels genotyped in the Illumina data and compared them between platforms. For each structural variant, the ONT genotypes were assumed to be true positives, and all genotype calls were binarised into presence (1) or absence (0). For each structural variant, and given that Y chromosomes are hemizygous, homozygous and heterozygous alternative calls were considered present, and homozygous reference genotypes absent.

As we wanted to study the correlation of binary variables, we made use of phi coefficients, also known as Matthews correlation coefficient or MCC. Phi coefficients should be interpreted similarly to a Pearson correlation coefficient. For the *1kgp* data, we considered a structural variant to be present if it was at a frequency higher than 0.2.

**Studying methylation using ONT**. The methylation status was called using *nanopolish v0.13.2*[46], which assigns a log-likelihood ratio to each individual CpG site. To avoid adding noise to the methylation results, we only used reads with the highest mapping quality as provided by minimap2 (mapQ = 60) and filtered out all others. We used the default log-likelihood threshold of 2 as implemented in *nanopolish v0.12* onward. As suggested by the developers, we called methylation with the option *--min-separation 5* to help calling CpG dense regions. The methylation frequency was calculated for each site as the number of mapped reads predicted as methylated divided by the number of total mapped reads.

We filtered out the few instances in which alternative alleles were present in a genomic position with cytosine in the reference sequence. We performed quantile normalization on the methylation values across samples with a minimum coverage of 4x using the R package *preprocessCore v1.56.0*[77]. CpG and gene annotations were obtained using the R package *annotatr v1.24.0*[78]. Minor modifications were made to these annotations for different analyses. All these modifications have been specifically described when used in the text. For the overlapping regions in the gene annotations, the priority set was the following: promoters, UTRs (5', 3'), first exon, non-first exons, all introns, and upstream region.

**Reporting summary**. Further information on research design is available in the Nature Portfolio Reporting Summary linked to this article.

## Data availability

All sequencing data generated for this study have been deposited at the European Nucleotide Archive (ENA) under the study accession PRJEB58141. Assemblies are deposited at the ENA under the study accession PRJEB59245. Raw sequencing data for the A0 haplogroup (cell line HG02982) were retrieved from the ENA study accession PRJEB28143 and its assembly from the accession ULGL01000000. The source data underlying Figs. 1–3 are provided in Supplementary Data 8. All other relevant data are available upon request.

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

## Acknowledgements

M.P.-F. has the support of an INPhINIT Retaining Fellowship from "La Caixa" Foundation (ID 100010434) with code LCF/BQ/DR20/11790032. T.M.-B. is supported by funding from the European Research Council (ERC) under the European Union's Horizon 2020 research and innovation programme (grant agreement No. 864203), PID2021-126004NB-100 (MICIIN/FEDER, UE) and Secretaria d'Universitats i Recerca and CERCA Programme del Departament d'Economia i Coneixement de la Generalitat de Catalunya (GRC 2021 SGR 00177).

## Author contributions

T.M.-B. and L.F.K.K. conceived the study. O.F., E.J., and E.R. cultured cells and performed the flow cytometry. A.F. cultured cells. P.E.-C., C.F., and L.L. performed adaptive sampling. I.G. and J.H. performed Illumina sequencing. L.F.K.K., A.S.-A., L.F.-P., and E.L. provided analytical support. M.D. and M.T. helped with data curation and analyses. M.P.-F. generated assemblies and structural variant calls. P.E.-C. performed methylation analyses, supervised all other analyses and generated most figures, including all main figures. D.J. designed and supervised analyses. P.E.-C. and M.P.-F. wrote the manuscript with input from all co-authors.

## Competing interests

L.F.K.K. is currently an employee of Illumina Inc. All other authors declare no competing interests.
