## [Peer Review File · Communications Biology]

Reviewers' comments:

Reviewer #1 (Remarks to the Author):

This paper uses nanopore sequencing to assemble the Y chromosome from 7 haplogroups. They nicely compare chromosome sorting to adaptive sampling and show that the latter provide equivalent data while being substantially more experimentally feasible. They go on to show the benefits of nanopore sequencing for structural variation detection on the Y chromosome, and even for DNA methylation analysis. Overall, this is well-presented, clear and convincing analysis that demonstrates several useful technical and biological insights.

Can the authors comment on the more complete Y chromosome sequence available through the telomere-to-telomere genome assembly (<https://www.biorxiv.org/content/10.1101/2022.12.01.518724v1>)? Would any of their results change if they aligned their reads or compared their result to this assembly? Would any of the structural variants be different if compared to this assembly? Although the authors may have done their analyses before this T2T Y chromosome reference was available, the paper would benefit and will have a greater impact if some of the analyses are updated to this newer assembly, at least considered with respect to it.

Another limitation is the use of LCLs, and one per haplogroup. It is possible that some of the structural variants they detect arose in vitro during LCL culturing. This could be controlled for by using one or more additional cell lines from the same haplogroup. For the least, the authors should comment on this caveat.

Reviewer #2 (Remarks to the Author):

This paper presents a technology-centric analysis of approaches to resequence and analyze Y chromosomes in humans. They employed Oxford Nanopore sequencing methods and genome assemblies for 7 cell lines representing 7 major human Y chromosome haplogroups. They used both adaptive sampling and flow cytometry to enrich the sample for Y chromosome sequences, and much of the focus of the paper was the comparison of these two approaches. The methods clearly explain how adaptive sampling works, and many readers may be unfamiliar with this feature of nanopore sequencing (after sequencing a few hundred basepairs, the fragment may be ejected if it does not show a match to collection of Y reference sequences). A couple sentences explaining this in the main text would be helpful.

Regarding the comparison of the two enrichment methods, the results are not very surprising but still are illuminating. Given the ease of doing adaptive sampling, it was especially interesting to see how well it performed. Its major drawback was that its reliance on mapping to a reference, heavily biasing the results against large insertions relative to the reference. Chromosome sorting, on the other hand is slow and expensive, and suffers from co-isolation of chromosome 22. But ultimately the best method will be to perform de novo assembly of each individual's Y chromosome, and then to perform the contrast with software method such as Assemblytics. For example, the adaptive sampling approach failed to find a 50 kb insertion relative to the reference, whereas this was easy to spot with the assembly-to-assembly contrast. The paper establishes this clearly. Valuable insight is also given on the pros and cons of Sniffles software for inference of structural variants (this too relies on contrast to a reference).

One issue that needs to be fixed is the failure to pay any attention to the composition of the indels. Surely many were transposable elements. Addition of this annotation would add an interesting and under-studied attribute of Y chromosome polymorphism.

Results of the Nanopolish analysis, identifying methylation status from the dwell time in Nanopore sequence runs, was a bit underwhelming. There were clearly differences along the Y chromosome in methylation status, most of which made biological sense, but only fairly drastic differences among haplogroups could be detected. Even in these cases there was no definitive association of methylation status with differences in gene expression or other attributes of gene function.

Reviewer #3 (Remarks to the Author):

This is an interesting part methods validation/ part finding study on sequencing of the Y chromosome. Sequencing of the Y has been plagued for many years with methodological deficiency. The current authors describe, validate and contrast adaptive sampling and flow cytometry chromosome sorting. Their findings are well described in terms of the sequence generated. They validate adaptive sampling. This is makes it much easier for other scientists to sequence sections they are interested in. The epigenetic section is slightly less convincing and deserves a little bit more attention in the introduction and findings. Nonetheless I find this study well written and a valid new addition to the literature on the Y and sequencing. I would have liked to see a figure summarising the methodology and techniques used for the reader. Conservative filtering as described remains a problem maybe the authors want to suggest a way forward with this technology. This study will be of interest to the Y community of other species including animal models as well.

Reviewer #1 (Remarks to the Author):

This paper uses nanopore sequencing to assemble the Y chromosome from 7 haplogroups. They nicely compare chromosome sorting to adaptive sampling and show that the latter provide equivalent data while being substantially more experimentally feasible. They go on to show the benefits of nanopore sequencing for structural variation detection on the Y chromosome, and even for DNA methylation analysis. Overall, this is well-presented, clear and convincing analysis that demonstrates several useful technical and biological insights.

>>>We thank the Reviewer for their positive comments regarding our paper.

Can the authors comment on the more complete Y chromosome sequence available through the telomere-to-telomere genome assembly

(<https://www.biorxiv.org/content/10.1101/2022.12.01.518724v1>)?

Would any of their results change if they aligned their reads or compared their result to this assembly? Would any of the structural variants be different if compared to this assembly? Although the authors may have done their analyses before this T2T Y chromosome reference was available, the paper would benefit and will have a greater impact if some of the analyses are updated to this newer assembly, at least considered with respect to it.

>>>We appreciate the observation of Reviewer 1 regarding the recent release of the Y chromosome T2T assembly, the manuscript of which was published only a few days after our manuscript had been sent for publication consideration.

The T2T Consortium generated a telomere-to-telomere (T2T) human Y chromosome assembly using multiple data types (two long-read sequencing technologies, chromatin ligation methods and optical mapping data). It is important to note that to achieve a T2T Y chromosome as presented, one needs to generate data for the full genome. Still, achieving this milestone is particularly impressive, especially considering the difficulty in assembling highly repetitive chromosomes, which often have collapsed regions due to duplicated and repetitive sequences in them. Nevertheless, we must

acknowledge the substantial amount of data generation, manual labour, and computational expenses required to produce this assembly, which are only available to a few labs worldwide. The use of high coverage of high-fidelity long reads of PacBio (HiFi) and ultra long ONT reads, in combination with the development of improved assembly algorithms that incorporate the complementary characteristics of these long reads (like the assembler Verkko)¹, appears to be the key to successfully assembling highly repetitive chromosomes or genomes. Nonetheless, the approaches followed in the present study are a much affordable alternative, which requires access to less data types but at the expense T2T chromosome assemblies.

In line with the Reviewer's suggestion, we inspected the extent to which our assemblies were comparable to the new T2T Y chromosome reference assembly. We observed that the assemblies presented in our study have less repetitive content than T2T assembly (Supplementary Table 4). This is mainly driven by the fact that the ampliconic region is not well resolved due to collapses in both our assemblies but also to the GRCh38 Y chromosome reference used in our study (Supplementary Fig. 8). As such, structural variants found in ampliconic regions might be attributed to distinct reads aligning to the same region, but originating from diverse sources. Because of that, we believe that using the T2T assembly as a reference can lead to improved detection and evaluation of structural variations in the ampliconic region.

In order to conform with Reviewer 1 suggestion, in the revised version of our manuscript we have now added one-to-one comparisons of the T2T Y chromosome assemblies with both the GRCh38 and one of the assemblies generated in this study –the haplogroup chosen is the one closest to the T2T Y chromosome reference haplogroup (Supplementary Table 4 and Supplementary Fig. 8). Moreover, we have commented on the existence of this new T2T assembly and discussed how it could impact the results of some of our analyses, such as structural variant detection, specially in ampliconic regions.

One instance of a new addition regarding the recently published T2T Y chromosome reference is now included in the Results section, and reads as follows:

Page 7, line 177: *Recently, a complete T2T human Y chromosome assembly was published³⁷. This new chromosome assembly showed a higher proportion of repetitive elements (RE, Supplementary Table 4), most of which were satellite sequences. Namely 85% (53 Mb) of this new T2T-Y was annotated as repetitive³⁷, considerably more than the 31% (17.5 Mb) reported for GRCh38-Y and also the 68% (13-15 Mb) we found in the assemblies we generated in this study (Supplementary Table 4). Moreover, this assembly has resolved the ampliconic regions (Supplementary Fig. 8).*

Moreover, the new supplementary figure (Supplementary Fig. 8) comparing the new T2T assembly with GRCh38 and one of the assemblies generated in this study is displayed as follows:

Supplementary Figure 8. Dot-plot comparing Y chromosome assemblies. (A) T2T Y chromosome vs GRCh38 Y chromosome assembly and (B) T2T Y chromosome vs J haplogroup (HG03998 cell line) assembly manually scaffolded based on alignments to GRCh38. Segments are colored blue, red or green, representing forward, reverse or repetitive alignments, respectively.

Another limitation is the use of LCLs, and one per haplogroup. It is possible that some of the structural variants they detect arose in vitro

during LCL culturing. This could be controlled for by using one or more additional cell lines from the same haplogroup. For the least, the authors should comment on this caveat.

>>>We thank Reviewer 1 for highlighting this limitation and how it might affect our results regarding structural variants. We have commented on this caveat in the results and discussion sections of the revised manuscript (see below). However, it is also worth mentioning that one instance that could illustrate the structural stability of our cell lines, albeit a replicate, can be found in the assembly-to-assembly comparison of one of our cell lines with the T2T assembly (Supplementary Fig. 8B). We only observe a small number of regions present in our assembly and absent in the T2T, thus suggesting structural stability in our cell line.

New text added to the manuscript highlighted in yellow:

Results:

Page 9, line 267: *As expected, intra-haplogroup associations were typically positive and significant, generally having stronger associations than inter-haplogroups comparisons (Supplementary Fig. 13). The use of additional cell line replicates for each haplogroup might have yielded higher associations, as it would have helped discard instances of variation that arose during immortalization.*

Discussion:

Page 15, line 426: *We also took advantage of the long-read data generated to explore the landscape of structural variants in each cell line. For that, we used two different methods for structural variant calling: one based on long-read mapping and another based on assembly comparison. The former allows for two rounds of genotyping and so the final candidates are potentially more curated. The latter is based on genome-to-genome comparisons, so it is able to detect longer genomics variants. We consider that for low data samples the creation of a de novo Y chromosome assembly may allow the detection of structural variants that cannot be recognized with a mapping-based method, considering the low coverage of reads mapping in those regions. Of note, we are aware that limited by only one replicate per haplogroup, we are not able to discriminate*

sample-specific variation than that appearing during the immortalization of these cells⁶¹. Moreover, the recent T2T Y chromosome³⁷ will allow for a more reliable and comprehensive assessment of the structural variants found in this study.

Reviewer #2 (Remarks to the Author):

This paper presents a technology-centric analysis of approaches to resequence and analyze Y chromosomes in humans. They employed Oxford Nanopore sequencing methods and genome assemblies for 7 cell lines representing 7 major human Y chromosome haplogroups. They used both adaptive sampling and flow cytometry to enrich the sample for Y chromosome sequences, and much of the focus of the paper was the comparison of these two approaches. The methods clearly explain how adaptive sampling works, and many readers may be unfamiliar with this feature of nanopore sequencing (after sequencing a few hundred basepairs, the fragment may be ejected if it does not show a match to collection of Y references sequences). A couple sentences explaining this in the main text would be helpful.

>>>We thank the Reviewer for their comments. Following Reviewer 2 suggestion, we have included a brief explanation on the rationale of adaptive sampling in the main text, which now reads as follows (highlighted in yellow):

*Page 2, line 60: Chromosome sorting allows the chromosome of interest to be sequenced on different platforms after its physical isolation by flow cytometry. This separation is possible because different chromosomes have specific fluorescence intensity¹⁹. On the other hand, adaptive sampling allows for the sequencing of specific DNA regions by **sequence-specific enrichment or depletion of off-target reads during sequencing** without the need for previous chromosome enrichment^{20,21}. **Any DNA molecule that does not correspond to the genomic region of interest will be ejected from the pore, thereby preventing any further sequencing.** To obtain a de novo assembly, it is also important to avoid whole genome amplification (WGA), as this process can introduce chimeras, bias the assembly process²² and prevent the detection of epigenetic modifications.*

Regarding the comparison of the two enrichment methods, the results are not very surprising but still are illuminating. Given the ease of doing adaptive sampling, it was especially interesting to see how well it performed. Its major drawback was that its reliance on mapping to a reference, heavily biasing the results against large insertions relative to the reference. Chromosome sorting, on the other hand is slow and expensive, and suffers from co-isolation of chromosome 22. But ultimately the best method will be to perform *de novo* assembly of each individual's Y chromosome, and then to perform the contrast with software method such as Assemblytics. For example, the adaptive sampling approach failed to find a 50 kb insertion relative to the reference, whereas this was easy to spot with the assembly-to-assembly contrast. The paper establishes this clearly. Valuable insight is also given on the pros and cons of Sniffles software for inference of structural variants (this too relies on contrast to a reference).

One issue that needs to be fixed is the failure to pay any attention to the composition of the indels. Surely many were transposable elements. Addition of this annotation would add an interesting and under-studied attribute of Y chromosome polymorphism.

>>>We agree with Reviewer 2 that a more thorough study of the composition of the indels could indeed enrich our understanding on the nature of the structural variant diversity in Y chromosomes. As such, we have now added new results regarding the annotation of the indels as provided by RepeatMasker² (see Methods' section *Indel composition*) in a new Supplementary Table (Supplementary Table 6). We have added these new findings to those describing the structural variants in the corresponding results section (highlighted in yellow below):

Page 7, line 201: *Out of the 801 variants found, there were 320 insertions and 481 deletions (including 3 insertions and 9 deletions from chrY_random). We then analyzed the repeat content of these variants and found that all indels from chrY_random consisted of LTR12B-like elements (from LTR retrotransposons type ERV1) (Supplementary Table 6). In the Y chromosome, we found 116 insertions and 153 deletions with repetitive elements (RE), and in the majority of these cases, the variants were almost entirely spanned by repeats (Supplementary Table 6).*

Next, we manually investigated the longest events detected using Sniffles and confirmed the longest deletion of 6,314 bp and longest insertion of 6,023 bp. Both were found in haplogroup A0 and belonged to different X-degenerate regions (Supplementary Figs. 9 and 10). The longest deletion had only 38% repeat content but included different types of repeats, such as ALU-like elements (AluSc and AluSx), L1-like elements (L1PA13), and Simple Repeats (CCTTn). The repeat content of the longest insertion (98.7% of the sequence) only comprised one single LINE L1-like element, L1HS, which is one of the few retrotransposition-competent human-specific L1s^{42,43} (Supplementary Table 6).

Even though the most common RE in indels are simple repeats for both insertions and deletions, proportionally more insertions were caused by smaller REs (for example ALUs or Satellite-like), and more deletions were detected in regions with long terminal repeats (LTRs), mostly ERV1 LTR12B-like elements (Supplementary Table 6). Interestingly, regions enriched with repeat arrays harboring LTR12B motifs have been previously reported as CNV hotspots⁴⁴.

Results of the Nanopolish analysis, identifying methylation status from the dwell time in Nanopore sequence runs, was a bit underwhelming. There were clearly differences along the Y chromosome in methylation status, most of which made biological sense, but only fairly drastic differences among haplogroups could be detected. Even in these cases there was no definitive association of methylation status with differences in gene expression or other attributes of gene function.

>>>We thank the Reviewer for their insightful comment on the methylation section. Our analyses are mainly limited by the lack of replicates per haplogroup. As such, we have restricted our results to highlighting particular instances of methylation differences (eg. Supplementary Fig. 24A), rather than extrapolating haplogroup-wise generalizations from our results or associating them with differences in gene expression. In this regard, we have now added a specific sentence clarifying the need for experimental validation in order to confirm any potential links between methylation and gene expression differences. See below, highlighted in yellow:

Page 12, line 336: *In that location, haplogroup A1a was found to be undermethylated compared to the other haplogroups, and although this difference was only modest, it could potentially modulate the expression levels of the gene located in this region. This gene is NLGN4Y, which is a long gene that spans over 300 kb and is expressed in brain and other tissues, including LCLs ($T_{NLGN4Y} = 0.714$). Interestingly, this gene has been proposed as a candidate for autism spectrum disorder^{57,58}. As expected, we found CGIs located upstream of this gene to be unmethylated (CGI_1 and CGI_2), which would be consistent with the expression of this gene in LCLs. On the contrary, a CGI potentially regulating an overlapping non-coding gene in the opposite strand and with no expression in LCLs was shown to be fully methylated in all cell lines (CGI_3). Experimental validation of the expression of these genes could ultimately confirm whether such differences in methylation truly affect gene expression levels or are merely coincidental.*

And in the discussion:

Page 16, line 448: *Still, methylation differences at the population level are expected to be small in magnitude⁴⁹. In this line, we were able to detect small differences in methylation in regions that could influence the regulation of specific genes. This is the case of gene NYGN4Y, which we found to fully overlap with a region with consistently lower methylation in the cell line belonging to haplogroup A1a. However, gene expression data from these cell lines could ultimately reveal whether such subtle variations in methylation translate into differences in gene expression. In the same line, the addition of biological replicates for different haplogroups could help uncover whether these differences are either cell line or haplogroup-specific.*

Moreover, and as it has been mentioned by the reviewer, our results are consistent with the methylation values we would expect to find in the different Y chromosome sequence classes and also conform with the different epigenetic and gene annotations examined. The consistency of these patterns shows the potential of nanopore to study methylation in structurally complex regions such as the Y chromosome, and we believe this is the main outcome of this section. As such, we have stressed this point in the results section.

Page 12, line 349: *Altogether, we show that ONT can be used to study 5mC across different cell lines, as it consistently recapitulates the main methylation patterns observed across mammalian genomes. Moreover, we also show that it can be helpful for the study of traditionally challenging genomic regions, particularly those present in the Y chromosome, although future work including replicates will clarify the actual sensitivity of this approach to detect differentially methylated regions.*

Reviewer #3 (Remarks to the Author):

This is an interesting part methods validation/ part finding study on sequencing of the Y chromosome. Sequencing of the Y has been plagued for many years with methodological deficiency. The current authors describe, validate and contrast adaptive sampling and flow cytometry chromosome sorting. Their findings are well described in terms of the sequence generated. They validate adaptive sampling. This makes it much easier for other scientists to sequence sections they are interested in. The epigenetic section is slightly less convincing and deserves a little bit more attention in the introduction and findings. Nonetheless I find this study well written and a valid new addition to the literature on the Y and sequencing.

>>>We thank the reviewer for their positive comments about our study. We agree that the epigenetic section could be better described. As such, we have expanded on the description of the rationale, main findings, and limitations in this particular section.

Some instances of these changes are found in the results section (highlighted below in yellow):

Page 11, lines 298: *These results alone already demonstrate the potential of ONT sequencing in studying DNA methylation in challenging genomic regions.*

Or at the end of this section:

Page 12, line 336: *In that location, haplogroup A1a was found to be undermethylated compared to the other haplogroups, and although this difference was only modest, it could potentially modulate the expression levels of the gene located in this region. This gene is NLGN4Y, which is a long gene that spans over 300 kb and is*

expressed in brain and other tissues, including LCLs ($T_{NLGN4Y} = 0.714$). Interestingly, this gene has been proposed as a candidate for autism spectrum disorder^{57,58}. As expected, we found CGIs located upstream of this gene to be unmethylated (CGI_1 and CGI_2), which would be consistent with the expression of this gene in LCLs. On the contrary, a CGI potentially regulating an overlapping non-coding gene in the opposite strand and with no expression in LCLs was shown to be fully methylated in all cell lines (CGI_3). Experimental validation of the expression of these genes could ultimately confirm whether such differences in methylation truly affect gene expression levels or are merely coincidental.

Altogether, we show that ONT can be used to study 5mC across different cell lines, as it consistently recapitulates the main methylation patterns observed across mammalian genomes. Moreover, we also show that it can be helpful for the study of traditionally challenging genomic regions, particularly those present in the Y chromosome, although future work including replicates will clarify the actual sensitivity of this approach to detect differentially methylated regions.

I would have liked to see a figure summarizing the methodology and techniques used for the reader.

>>>Following the Reviewer's suggestion, we have now added a panel in Figure 1 (Fig. 1A) summarizing our experimental setup and the analyses used in this project. This way, we believe it will help the reader better understand how this project is organized.

Fig. 1. Study design, enrichments, and assemblies. (A) Summary of the samples, methodologies and analyses used in the study. (B) Phylogenetic tree of the human Y chromosomes used in the study. Split times taken from Jobling & Tyler-Smith, 2017³². kya, kilo years ago. (C) Enrichment factor values of the H haplogroup from data generated using chromosome sorting and adaptive sampling. The chrY shows higher enrichment with chromosome sorting than with adaptive sampling for the haplogroup compared. The dashed vertical line equal to 1 denotes no chromosomal enrichment. (D) Dot-plots of the manually scaffolded Y chromosomes compared to the resolved MSY region of GRCh38. The large-scale deletion in the J haplogroup is most likely due to its low coverage.

Conservative filtering as described remains a problem maybe the authors want to suggest a way forward with this technology. This study will be of interest to the Y community of other species including animal models as well.

>>>We agree with the Reviewer that our conservative filtering is a major limitation, and it was mainly due to the lack of biological replicates. However, as long-read sequencing technologies achieve longer and more accurate reads, direct *de novo* assemblies of highly repetitive chromosomes will become more feasible. Because of this, we expect downstream analyses that rely on these assemblies, such as the identification of structural variants, to be more reliable and comprehensive. As suggested by Reviewer 1, we have included some lines discussing the limitations of not having biological replicates. We also added a concluding remark with our thoughts on the future of long-read technologies.

Page 16, line 463: Taken together, here we provide a framework to study complex genomic regions. We applied this simple, fast and affordable technology to study diverse human population groups. Moreover, this approach can be applied to the generation of long-read data of other regions or chromosomes of interest. Long-read technologies are achieving longer and more accurate reads, which, together with higher throughputs, will enable more reliable and comprehensive comparative genomics studies, particularly for chromosomes that contain high repeat content. As such, it could be used for the characterization of virtually any species, although it would be especially advantageous for those rich in complex genomic features.

References

1. Rautiainen, M. et al. Telomere-to-telomere assembly of diploid chromosomes with Verkko. Nat. Biotechnol. (2023) doi:10.1038/s41587-023-01662-6.
2. Smit, A. F. A., Hubley, R. & Green, P. RepeatMasker Open-4.0. 2013--2015. Preprint at (2015).

REVIEWERS' COMMENTS:

Reviewer #1 (Remarks to the Author):

The authors have addressed my concerns.

Reviewer #2 (Remarks to the Author):

Apologies for my delayed re-review, but obligations arose beyond my control. I finally had a chance to read the reviews and authors' responses, along with the revised MS, and the authors have done a nice job responding. Addition of the comparisons to the T2T assembly are especially informative and improve the paper greatly. Some of the other responses mostly appear as softening of conclusions, but the main punchline of the paper remains intact. I conclude that the revised manuscript now satisfies the issues raised by my review (and the others too, as far as I am concerned).

Reviewer #3 (Remarks to the Author):

The authors has answered all
My queries.